# Interaction of TLK1 and AKTIP as a Potential Regulator of AKT Activation in Castration-Resistant Prostate Cancer Progression

Md Imtiaz Khalil [1],[†] , Christopher Madere [1],[†] , Ishita Ghosh [1] , Rosalyn M. Adam [2] and Arrigo De Benedetti [1],*

1   LSU Health Sciences Center, Department of Biochemistry and Molecular Biology, Shreveport, LA 71103, USA; mdimtiaz.khalil@lsuhs.edu (M.I.K.); ckmadere@yahoo.com (C.M.); Ishita.ghosh@lsuhs.edu (I.G.)
2   Department of Urology, Boston Children's Hospital and Department of Surgery, Harvard Medical School, Boston, MA 02115, USA; Rosalyn.Adam@childrens.harvard.edu
*   Correspondence: arrigo.debenedetti@lsuhs.edu; Tel.: +1-318-675-5668
†   These authors contributed equally.

**Abstract:** Prostate cancer (PCa) progression is characterized by the emergence of resistance to androgen deprivation therapy (ADT). AKT/PKB has been directly implicated in PCa progression, often due to the loss of PTEN and activation of PI3K>PDK1>AKT signaling. However, the regulatory network of AKT remains incompletely defined. Here, we describe the functional significance of AKTIP in PCa cell growth. AKTIP, identified in an interactome analysis as a substrate of TLK1B (that itself is elevated following ADT), enhances the association of AKT with PDK1 and its phosphorylation at T308 and S473. The interaction between TLK1 and AKTIP led to AKTIP phosphorylation at T22 and S237. The inactivation of TLK1 led to reduced AKT phosphorylation, which was potentiated with AKTIP knockdown. The TLK1 inhibitor J54 inhibited the growth of the LNCaP cells attributed to reduced AKT activation. However, LNCaP cells that expressed constitutively active, membrane-enriched Myr-AKT (which is expected to be active, even in the absence of AKTIP) were also growth-inhibited with J54. This suggested that other pathways (like TLK1>NEK1>YAP) regulating proliferation are also suppressed and can mediate growth inhibition, despite compensation by Myr-AKT. Nonetheless, further investigation of the potential role of TLK1>AKTIP>AKT in suppressing apoptosis, and conversely its reversal with J54, is warranted.

**Keywords:** TLK1; AKTIP/FT1/FTS; PKB/AKT; prostate cancer; castration-resistant prostate cancer; apoptosis

## 1. Introduction

Prostate cancer (PCa) is one of the most frequently diagnosed malignancies in men in the United States. The disease becomes lethal when patients stop responding to androgen deprivation therapy (ADT). Since PCa cells require androgen receptor (AR) signaling for their growth and survival, androgen withdrawal results in the regression of prostate tumors. However, tumors will relapse invariably within 2–3 years and progress to the more aggressive 'castration-resistant' phenotype. Deregulation of the balance between the proliferation and apoptotic rates of tumor cells contributes to much of the formation of prostatic adenocarcinoma; however, castration-resistant prostate cancer (CRPC) is mainly a consequence of the apoptotic suppression of the cells (reviewed in [1,2]). Androgen refractory cells within the tumor's microenvironment may initiate signaling cascades that promote the survival of the cells and shift the equilibrium more toward cell proliferation. Therefore, targeting the molecular factors inducing the anti-apoptotic response could be an excellent treatment strategy for CRPC.

Tousled-like kinase 1 (TLK1), a mammalian homologue of plant tousled kinase (TSL), was first cloned by the Niggs group in 1999 (and later) and was reported to play roles in chromatin maintenance, i.e., replication, transcription, DNA damage response, and repair [3–9]. Findings from our lab reported the TLK1 role in prostate tumorigenesis through

the activation of the DNA damage response [10], the apoptotic prevention regulating mitochondrial membrane permeability [11], and recently, through YAP stabilization by activating and interacting with NEK1 [12]. In addition, androgen withdrawal is reported to increase the expression of TLK1B through the compensatory activation of mTORC1 [10]. Like other kinases, TLK1 may also have a broad range of substrate specificity. To determine TLK1 interactomes, a novel proteomic screen was conducted by our lab, which identified 165 full-length human proteins that interact with TLK1 [13]. The AKT-interacting protein (AKTIP), also called FT1 or FTS, was found to interact with TLK1 with high confidence.

FT1 was first discovered as a deletion mutant in mice that led to abnormalities in limb development, thymic hyperplasia, and the reduced apoptotic response of thymocytes to dexamethasone treatment [14,15]. The double-deletion mutant of AKTIP resulted in embryogenic lethality in mice at around the mid-gestation period, with severe malformation of the fore-and midbrain regions [14]. The human homologue of FT1 shares about 96% sequence homology with mice, which suggests the functional conservation of this protein between the two species. AKTIP shares homology with an E2 ubiquitin-conjugating enzyme variant, however, it lacks a cysteine residue, to which ubiquitin is covalently bound [14]. The reported interacting proteins of AKTIP included protein kinase B/AKT [16], PDK1 [16], shelterin complex proteins TRF1 and TRF2 [17], lamin A and B [18], Notch1 and Hes1 [19], and EGFR [20]. AKTIP directly binds to AKT and PDK1 and enhances the phosphorylation of AKT on both of its regulatory sites (T308 and S473) by promoting its interaction with upstream kinase PDK1, and thereby providing a better scaffold for these kinases. PKB/AKT is a serine/threonine kinase which is vital for the control of a multitude of cellular processes from cell growth and proliferation to cell survival and anti-apoptotic response (reviewed in [21–23]). Upon growth-related stimulation, PI3K catalyzes the conversion of PIP2 to PIP3, which anchors the pleckstrin homology (PH) domain-containing proteins, such as AKT and PDK1. Upon association, AKT is phosphorylated on T308 by PDK1, followed by the phosphorylation on S473 by mTORC2 complex, leading to the full activation of AKT (reviewed in [24,25]). The deregulation of AKT activity is evident in many different oncogenic activities, especially in CRPC [2,26,27]; therefore, the objective of this study is to elucidate if TLK1-AKTIP signaling may regulate AKT activation and subsequent PCa cell survival. Our study suggests that TLK1 can directly bind and phosphorylate AKTIP. Both the pharmacologic and genetic depletion of TLK1 and AKTIP lead to the disruption of phosphorylation on the AKT regulatory sites and, hence, impair PCa cell survival.

## 2. Materials and Methods

### 2.1. Plasmids and Antibodies

The wild-type, full-length human TLK1B bacterial expression plasmid was obtained from Dr. Sivapriya Kirubakaran, Discipline of Biological Engineering, Indian Institute of Technology, Gandhinagar, India [28]. The wild-type full-length human GST-AKTIP bacterial expression plasmid was a kind gift from Dr. Romina Burla and Dr. Isabella Saggio, Dipartimento di Biologia e Biotecnologie, Sapienza, Università di Roma, Roma, Italy [17]. The following primary antibodies were used in this study: rabbit anti-TLK1 (Thermo Fisher, Waltham, MA, USA, cat# 720397), rabbit anti-TLK1B (lab-generated), rabbit anti-AKTIP (Thermo Fisher, Waltham, MA, USA, cat# PA5-61209), rabbit anti-AKT (Cell Signaling Technology, CST, Danvers, MA, USA, cat# 9272), rabbit anti-phospho-AKT (S473) (CST, Danvers, MA, USA, cat# 9271), rabbit anti-phospho-AKT (T308) (CST, Danvers, MA, USA, cat# 9275), rabbit anti-GAPDH (CST, Danvers, MA, USA, cat# 2118S), and HRP-conjugated anti-β-tubulin (Santa Cruz Biotechnology, SCBT, Dallas, TX, USA, cat# sc-23949).

### 2.2. Cell Culture

HEK 293, HeLa, and LNCaP cells were purchased from American Type Culture Collection (ATCC, Manassas, VA, USA) and authenticated within the past three years. HEK 293 and HeLa cells were cultured in DMEM media, supplemented with 10% fetal

calf serum (FCS) and 1% penicillin/streptomycin. LNCaP cells were cultured in RPMI 1640 media, supplemented with 1% penicillin/streptomycin and either 10% FCS or 10% charcoal strip serum (CSS). All the cells were maintained in a humidified incubator at 37 °C with 5% $CO_2$.

### 2.3. Cell Treatment

HEK 293 or LNCaP cells were plated as $5 \times 10^5$ cells in each well of a 6-well plate and grown until 70–80% confluency. Cells were treated with 10 µM of either thioridazine (THD; Sigma Aldrich, St. Louis, MO, USA, cat# T9025) (or J54) [29] or increasing concentrations of THD (or J54) (0, 1, 5, 10, 15 & 20 µM) for 24 hrs. After the treatment, the cells were harvested for western blotting analysis.

### 2.4. Cell Transfection

HEK 293 or LNCaP cells were plated as $4 \times 10^5$ cells/well in a 6-well plate and grown until the cells became 70–80% confluent. The knockdown of AKTIP was carried out using 10 nM of AKTIP specific siRNA (Thermo Scientific, Waltham, MA, USA, cat# 4392420) by a Lipofectamine 3000 (Thermo Scientific, Waltham, MA, USA, cat# L3000-015) transfection reagent, following manufacturer's protocol, for 48 h. After the transfection, cells were harvested for WB analysis. The shRNA-mediated knockdown of TLK1 was described by Khalil et al. (2020) [12].

### 2.5. His Pull-Down Assay

Full-length, wild-type recombinant His-tagged TLK1B, wild-type GST-tagged AKTIP, T22A GST-AKTIP, and S237A GST-AKTIP proteins were purified, according to the published literature [17,28]. His-TLK1B was incubated with an equal amount of GST-AKTIP or 400 µg of HEK 293 cell lysates, and incubated on ice for 30 min. Additionally, 30 µL of pre-equilibrated Ni-NTA agarose beads (50% slurry) were added to the reaction, and the mixture was incubated on ice for another 30 min. The beads were spun down and washed three times with the cell lysis buffer. Bound proteins were eluted using a 1X Laemmli buffer, and the AKTIP protein was detected by WB analysis.

### 2.6. Co-Immunoprecipitation (Co-IP)

A total of 200 µg of HEK 293 or LNCaP cell lysate was incubated, either with control mouse IgG or AKTIP specific antibody for 4 h on ice. Additionally, 50 µL of preequilibrated protein A/G agarose beads (50% slurry) (SCBT, Dallas, TX, USA, cat# sc-2003) was added to the reaction, and the mixture was rotated overnight at 4 °C. After the incubation, the beads were spun down, washed three times with lysis buffer, and eluted by 1X Laemmli buffer. The AKTIP protein was detected by WB analysis from the elution content.

### 2.7. Site-Directed Mutagenesis

Site-directed mutagenesis (SDM) was performed using a QuikChange Lightning Site-Directed Mutagenesis Kit (Agilent Technology, Santa Clara, CA, USA, cat# 210519), following manufacturer's protocol. The sequences of the primers used to generate the GST-AKTIP mutant were: T22A_Mutation Primer (5-GGTGAAGAGAGGGCATTAACAGGGGACGTGAAAACC-3') and S237A_Mutation Primer (5'-CCCTATGCAATTGCCTTTTCTCCATGGAATCCTTCTG-3').

### 2.8. In Vitro Kinase (IVK) Assay

Both radioactive and non-radioactive IVK assays were conducted, as previously described [12]. Briefly, the recombinant purified GST-AKTIP protein was incubated with or without His-TLK1B, along with kinase buffer containing non-radioactive ATP. The samples were run in an SDS-PAGE gel and GST-AKTIP bands were excised for mass spectrometric analysis. For radioactive IVK assay, His-TLK1B was incubated with wild-type GST-AKTIP, T22A GST-AKTIP, or S237A GST-AKTIP protein, along with a kinase buffer and [$\gamma$-$^{32}$P]

ATP. The samples were run in an SDS-PAGE gel, and radiation exposure was captured in an x-ray film.

### 2.9. Identification of AKTIP Phospho-Peptides

The identification of the phosphoresidues of AKTIP by TLK1B was conducted at the University of Kentucky, Proteomics Core Facility, as previously described [12].

### 2.10. Cell Proliferation Assay

The cell proliferation assay was conducted, as previously described [30]. Briefly, 2000 cells/well were seeded in a 96-well plate, with 10% FCS containing media with or without different concentrations of J54. The plate was incubated in the IncuCyte zoom incubator at 37 °C with 5% $CO_2$ for 96 h. Images were taken by the IncuCyte (Essen Bioscience, Ann Arbor, MI, USA) at every 4 h, and the cell occupancy percentage over time was determined.

### 2.11. Western Blotting

Cell lysates were prepared in a 1X RIPA lysis buffer (SCBT, Dallas, TX, USA, cat# 24948) by sonication, and the protein concentration was determined using a BCA protein assay kit (Thermo Scientific, Waltham, MA, USA, cat# 23225). The samples were run in a SDS-PAGE gel and transferred to a PVDF membrane. The blocking of the membrane was conducted in 5% non-fat, dry milk. After washing, the membrane was incubated in primary antibodies overnight in a rocker at 4 °C. After washing three times, the blots were incubated in HRP-conjugated secondary antibodies at room temperature for 1 h and subsequently incubated in ECL substrates (Thermo Scientific, Waltham, MA, USA, cat# 32106). The chemiluminescent detection of the protein bands was conducted using a Biorad ChemiDoc imaging system (Biorad, Hercules, CA, USA, cat# 12003154).

## 3. Results

### 3.1. TLK1 Interactions with AKTIP Both In Vitro and In Vivo

Proteomic screening from our lab identified AKTIP as an interacting protein of TLK1B in vitro [13]. We further confirmed their in vitro interaction by incubating purified recombinant His-tagged TLK1B and GST tagged AKTIP. The Ni-NTA agarose pull-down of His-tagged TLK1B also precipitated GST-tagged AKTIP, which confirms their direct and physical interaction in vitro (Figure 1a and its replica in Figure S1A). To test if TLK1 can interact with AKTIP in cells, we conducted another His pull-down assay, incubating purified recombinant His-tagged TLK1B with HEK293 cell lysates, which demonstrated that recombinant TLK1 can interact strongly with endogenous AKTIP (Figure 1b). Finally, we conducted a co-IP assay using HEK293 and LNCaP cell lysates, incubated with TLK1 specific antibodies. Although, AKTIP was precipitated with TLK1 in both cell lines, the TLK1-AKTIP interaction was shown to be stronger in HEK293, compared to LNCaP cells (Figure 1c). The expression of AKTIP and TLK1 in the two cell lines is similar (see Figure 3 and Figure S1B). We also investigated the interaction of AKTIP with AKT and whether its potential phosphorylation by TLK1 could affect their association. We found that the inhibition of TLK1 activity with THD did not affect the amount of AKT that was co-immunoprecipitated with AKTIP (Figure 1d).

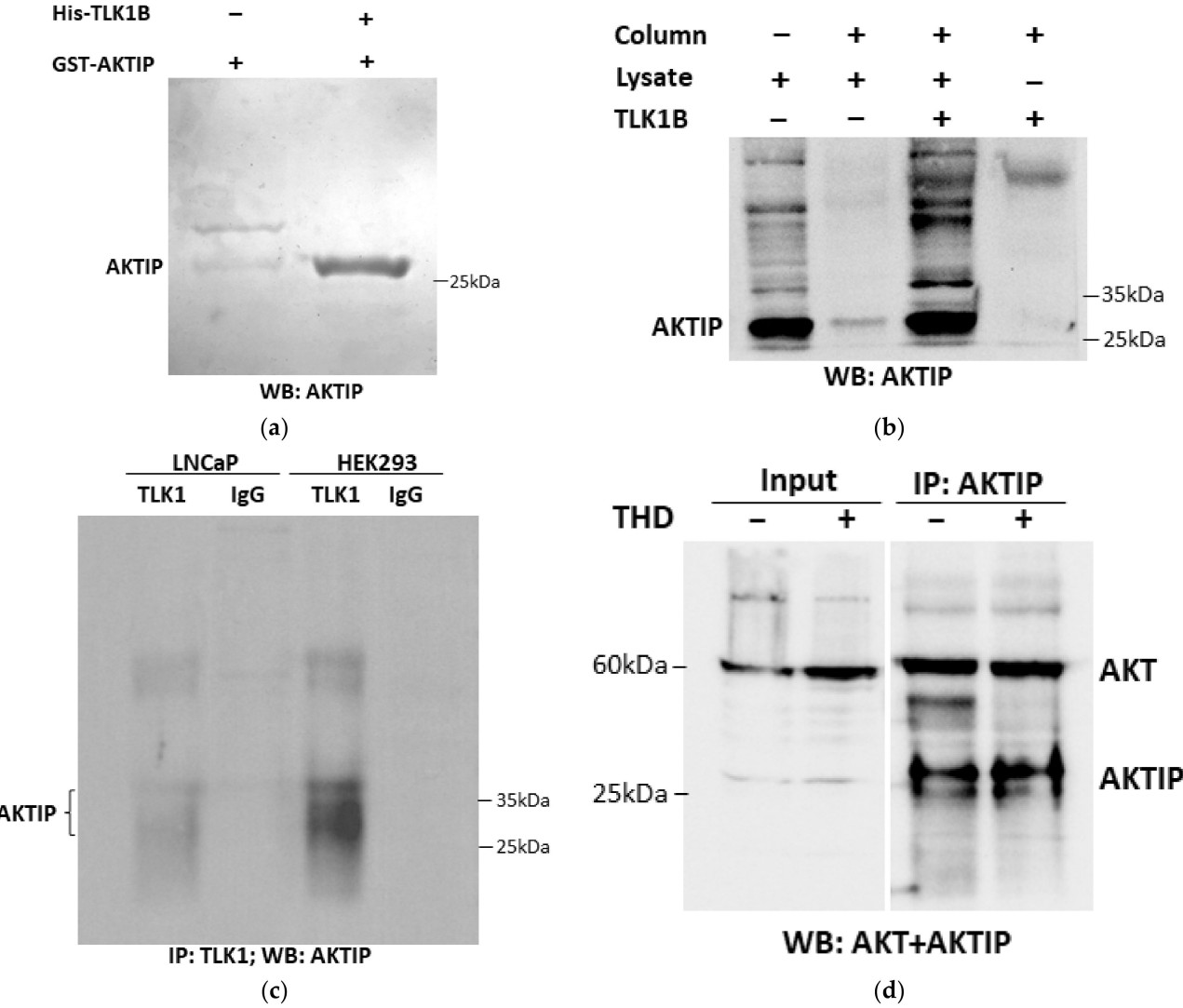

**Figure 1.** Interaction of TLK1 and AKTIP. (**a**) In vitro pull-down of recombinant His-TLK1B and GST-AKTIP, isolated on Ni-NTA Sepharose; 1 µg of each protein were allowed to bind to each other and to the beads, followed by extensive washing and WB for AKTIP. (**b**) Ni-NTA Sepharose pull-down of endogenous AKTIP from HEK 293 cell extract, incubated with or without recombinant His-TLK1B. AKTIP pull-down was determined by WB. (**c**) Co-IP of TLK1 and AKTIP form extracts of LNCaP and HEK 293 cells using TLK1 antiserum, followed by WB for AKTIP. (**d**) Co-IP of AKTIP using AKTIP specific antibody from LNCaP cell extracts with or without THD treatment and sequential immunoblotting of AKT and AKTIP.

*3.2. TLK1 Phosphorylates AKTIP In Two Unique Residues, Which Lie in the Intrinsically Disordered Region*

Because kinase-substrate interaction often leads to the phosphorylation of the substrate, we tested whether TLK1 can phosphorylate AKTIP. A non-radioactive in vitro kinase assay was conducted using a purified recombinant His-tagged TLK1B and purified recombinant GST-tagged AKTIP. The determination of AKTIP phosphopeptides by mass spectrometry was conducted at the University of Kentucky's proteomics facility. Both control samples (AKTIP and TLK1B+ AKTIP) were subjected to dithiothreitol reduction, iodoacetamide alkylation, and in-gel trypsin digestion.

The resulting tryptic peptides were analyzed using an LTQ-Orbitrap mass spectrometer. The AKTIP protein was detected with a high-protein score and good coverage in both samples (control AKTIP = 60.27% and TLK1B + AKTIP = 65.07%). While no phosphorylation sites were detected in the control AKTIP samples purified from bacteria, the TLK1B + AKTIP samples possessed several phosphorylated residues (Figure 2a).

Two phosphopeptides ($S_{16}$EGEEKTLTGDVK$_{28}$ and $I_{230}$EDPYAISFSPWNPSVHDEAR$_{252}$) each contained a phosphoresidue, identified by MASCOT software as T22 or T24 and S237 or S239 (Figure 2b). After manually inspecting the spectra, T22 and S237 are considered to be the likely TLK1 phosphorylation sites (Figure 2c). To check the authenticity of these phosphorylation sites, we conducted a radioactive IVK assay, using [$\gamma$-$^{32}$P] ATP as a radioactive source. The incubation of TLK1B with wild-type AKTIP, T22A, or S237A non-phosphorylatable variants resulted in the reduced phosphorylation signal compared to the wild-type AKTIP, as evident from the autoradiograph and densitometric quantification (Figure 2d,e). This suggests T22 and S237 as true target residues for phosphorylation by TLK1 on AKTIP. Finally, we mapped these phosphoresidues to the domains of AKTIP, which lie on the N- and C-terminals intrinsically disordered regions of AKTIP (Figure 2f).

## Control AKTIP

| Description | Score | Coverage |
|---|---|---|
| AKT-interacting protein OS=Homo sapiens OX=9606 GN=AKTIP PE=1 SV=1 - [AKTIP_HUMAN] | 1214.48 | 60.27 % |

**Found Modifications:**
C   Carbamidomethyl (C)
0   Oxidation (M)
P   Phospho_STY (S,T,Y)

|  | 1 | 11 | 21 | 31 | 41 | 51 | 61 | 71 | 81 | 91 |
|---|---|---|---|---|---|---|---|---|---|---|
| 1 | MNPFWSMSTS | SVRKRSEGEE | KTLTGDVKTS | PPRTAPKKQL | PSIPKNALPI | TKPTSPAPAA | QSTNGTHASY | GPFYLEYSLL | AEFTLVVKQK | LPGVYVQPSY |

101 — (C) — (O)

RSALMWFGVI FIRNGLYQDG VFKFTVYIPD NYPDGDCPRL VFDIPVFKPL VDPTSGELDV KRAFAKWRRN KNKIWQVLMY ARRVFYKIDT ASPLNPERAV

201

LYEKDIQLFK SKVVDSVKVC TARLFDQPKI EDPYAISFSP WNPSVHDEAR EKMLTQKKPE EQHNKSVNVA GLSWVKPGSV QPFSKEEKTV AT

## TLK1B+ AKTIP

| Description | Score | Coverage |
|---|---|---|
| AKT-interacting protein OS=Homo sapiens OX=9606 GN=AKTIP PE=1 SV=1 - [AKTIP_HUMAN] | 1676.11 | 65.07 % |

**Found Modifications:**
C   Carbamidomethyl (C)
0   Oxidation (M)
P   Phospho_STY (S,T,Y)

|  | 1 | 11 | 21 | 31 | 41 | 51 | 61 | 71 | 81 | 91 |
|---|---|---|---|---|---|---|---|---|---|---|
| 1 | MNPFWSMSTS | SVRKRSEGEE | KTLTGDVKTS | PPRTAPKKQL | PSIPKNALPI | TKPTSPAPAA | QSTNGTHASY | GPFYLEYSLL | AEFTLVVKQK | LPGVYVQPSY |

101 — (O) — (C) — (O)

RSALMWFGVI FIRNGLYQDG VFKFTVYIPD NYPDGDCPRL VFDIPVFKPL VDPTSGELDV KRAFAKWRRN KNKIWQVLMY ARRVFYKIDT ASPLNPERAV

201 — P (P) P — P

LYEKDIQLFK SKVVDSVKVC TARLFDQPKI EDPYAISFSP WNPSVHDEAR EKMLTQKKPE EQHNKSVNVA GLSWVKPGSV QPFSKEEKTV AT

(a)

**Figure 2.** *Cont.*

| AKTIP Kinase Reaction - Potential Phosphorylation sites | | |
|---|---|---|
| **Peptide** | **Potential Phosphorylation sites** | **MOWSE scores** |
| S$_{16}$EGEEK**T**LTGDVK$_{28}$ | **T22(Phosp)** | 42 |
| S$_{16}$EGEEKTL**T**GDVK$_{28}$ | **T24(Phosp)** | 42 |
| I$_{230}$EDPYAI**S**FSPWNPSVHDEAR$_{252}$ | **S237(Phosp)** | 40 |
| I$_{230}$EDPYAISF**S**PWNPSVHDEAR$_{252}$ | **S239(Phosp)** | 40 |

(**b**)

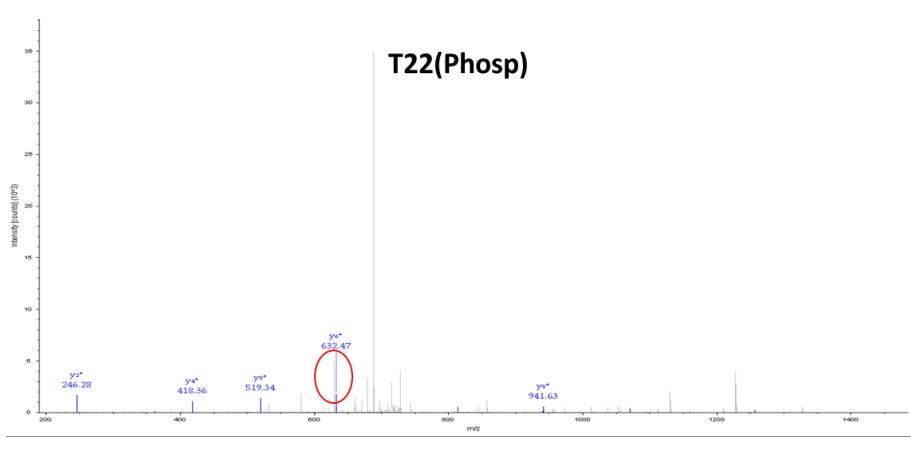

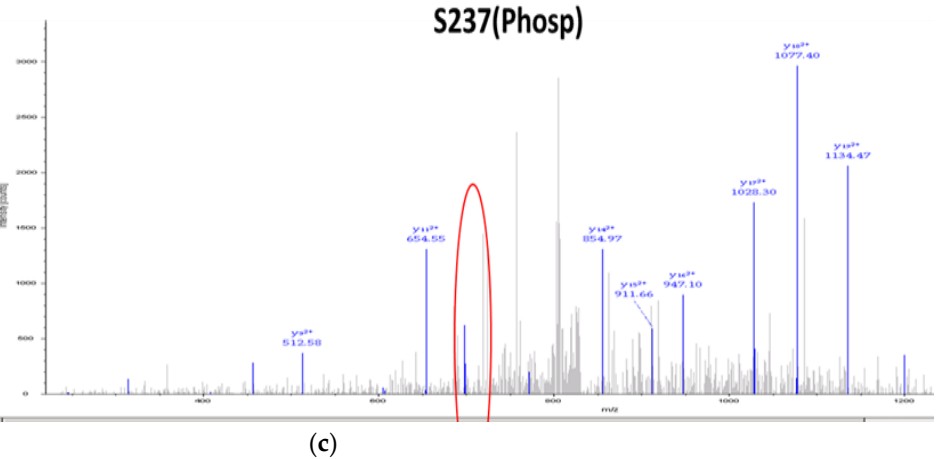

(**c**)

**Figure 2.** *Cont.*

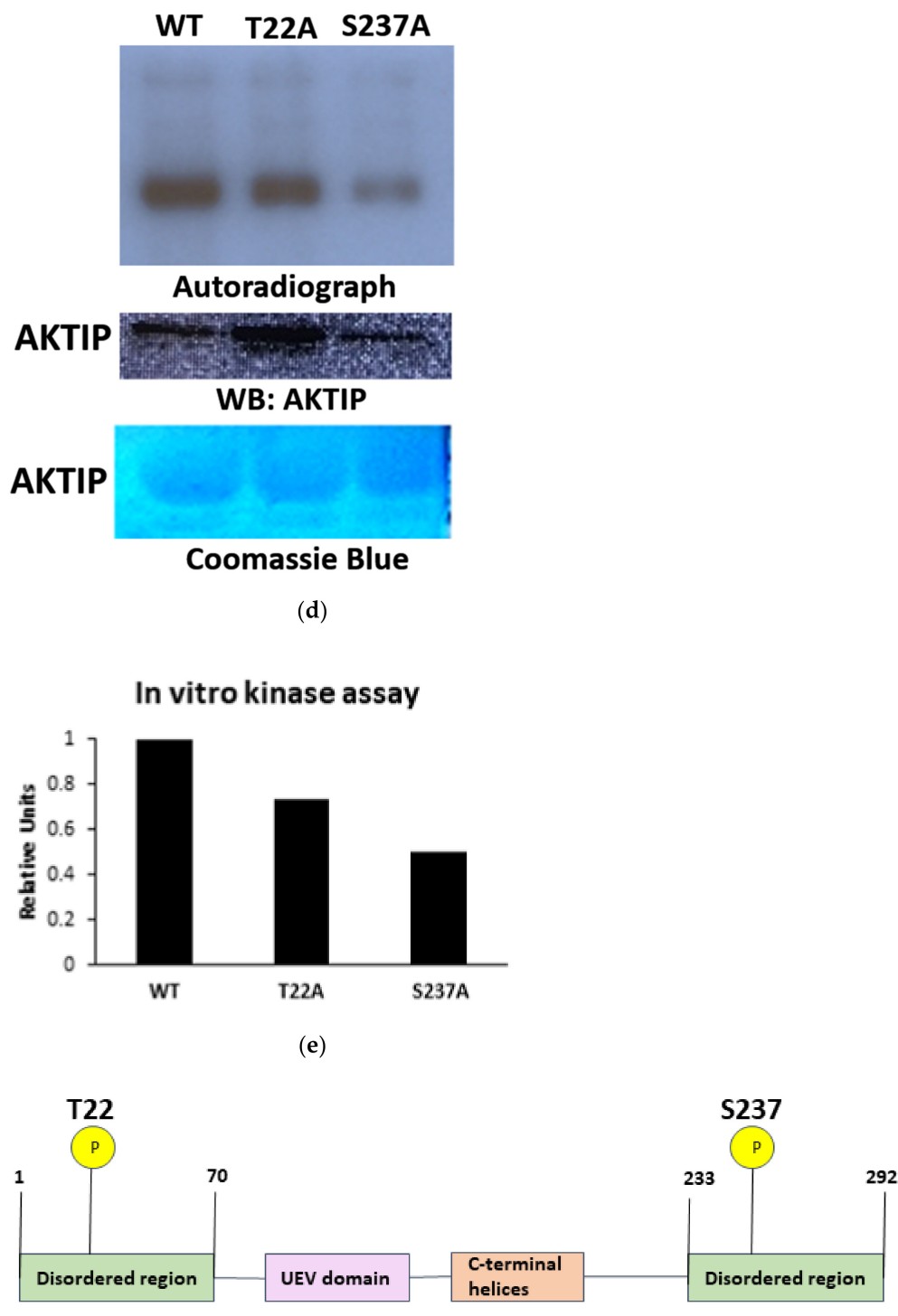

**Figure 2.** TLK1 phosphorylates AKTIP at T22 and S237. (**a**) AKTIP phosphopeptide determination by MS in control AKTIP (top panel) and TLK1B+ AKTIP samples (bottom panel). The unique phosphorylated residues in TLK1B+ AKTIP samples are marked by red circles. (**b**) Peptides containing unique phosphorylated sites detected in the TLK1B+AKTIP samples with their corresponding MOWSE scores. (**c**) Spectral analysis of AKTIP phosphoresidues T22 and S237 determination by MS. (**d,e**) In vitro phosphorylation of wild-type and mutant GST-AKTIP proteins by TLK1B and densitometric quantitation of the bands relative to amounts, as determined by WB. (**f**) Graphical outline of AKTIP domain structure and location of the phosphorylated residues.

### 3.3. The Genetic Depletion and Pharmacologic Inhibition of Both TLK1 and AKTIP Results in Reduced Activating Phosphorylation of AKT

To test whether the TLK1-mediated phosphorylation of AKTIP is important for the regulatory phosphorylation of AKT, we took both genetic and pharmacologic approaches to inhibit TLK1 and AKTIP. As mentioned earlier, ADT increases the expression of TLK1B, and consequently, LNCaP cells treated with charcoal-stripped serum (CSS) display higher AKT phosphorylation at two regulatory sites, i.e., T308 and S473, compared to cells treated with fetal calf serum (FCS) (Figure 3a,b).

As expected, the TLK1 inhibition by a small-molecule inhibitor (Thioridazine- THD) results in the reduction of phosphorylation in both the T308 and S473 sites of AKT, with a more robust reduction in FCS-treated cells. Similarly, the knockdown of AKTIP also results in reduced T308 AKT phosphorylation, with a higher effect on FCS-treated cells (Figure 3a,b). Consistent with the report that AKTIP overexpression increased AKT phosphorylation at both of its regulatory sites [16], we found that the siRNA-mediated knockdown of AKTIP decreased AKT phosphorylation at S473, and the effect is potentiated by the concomitant inhibition of TLK1 with THD, as evident from both HEK293 and LNCaP cells (Figure 3c,d). In addition, we tested a newly validated, small-molecule inhibitor of TLK1 (J54 [29]), in comparison to THD, and demonstrated that TLK1 inhibition results in reduced pAKT S473 levels, in a dose-dependent manner (Figure 3e,f). To further confirm that TLK1-mediated phosphorylation of AKTIP is essential for AKT phosphorylation, the shRNA-mediated knockdown of TLK1 was also conducted in HeLa cells, as described by Khalil et al., (2020) (Figure 3g) [12]. When the same blot was probed for pAKT S473, it showed that TLK1 depletion is associated with decreased AKT S473 phosphorylation (Figure 3g,h). We should stress that the amount of total AKT in all the WB in Figure 3 was not altered by either inhibitor TLK1 or siRNA to AKTIP.

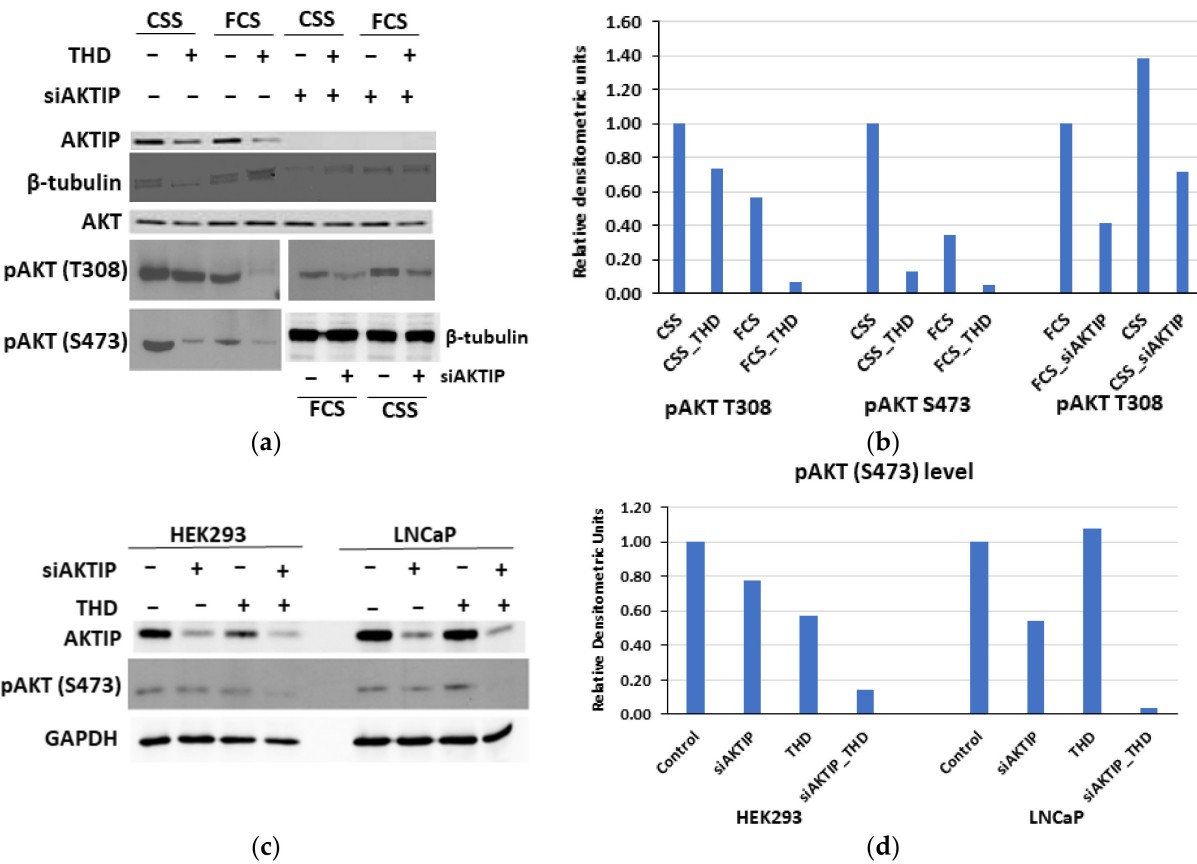

**Figure 3.** *Cont.*

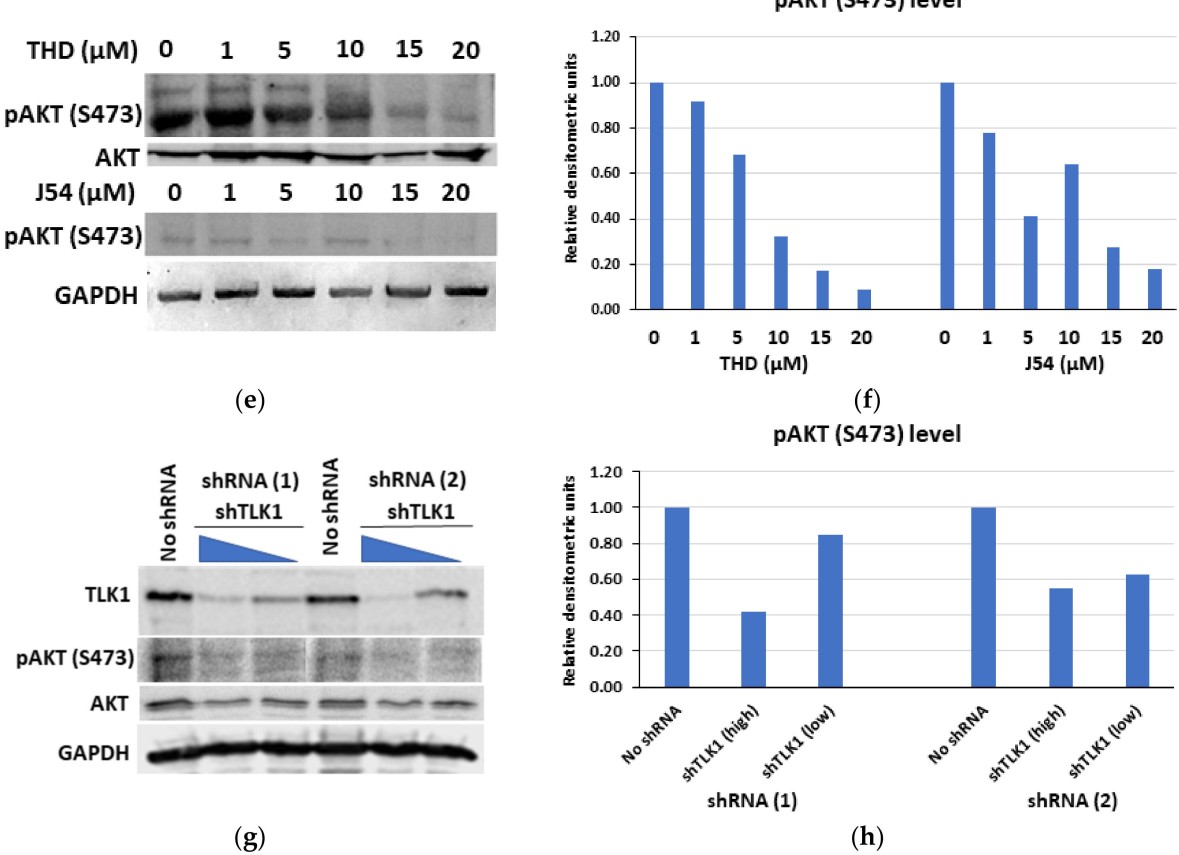

**Figure 3.** TLK1>AKTIP axis regulates activating AKT phosphorylations. (**a**) Incubation of LNCaP cell with charcoal-stripped serum (CSS), which elevates the expression of TLK1B, elevates the phosphorylation of AKT at T308 and S473, and this is suppressed with the TLK1 inhibitor Thioridazine (THD). Depletion of AKTIP largely suppresses AKT phosphorylation in both FCS and CSS conditions. (**c**) Depletion of AKTIP and addition of THD synergistically suppresses pAKT (S473) in both HEK 293 and LNCaP cells. (**e**) Treatment of LNCaP cells with two different inhibitors of TLK1 (THD or J54) results in dose-dependent loss of pAKT (S473). (**g**) Depletion of TLK1 with shRNA results in reduction of pAKT (S473). (**b,d,f,h**) Quantitation of the respective WBs.

*3.4. TLK1 Inhibition Reduces Cellular Proliferation Rate, Probably by Downregulating AKT Activation, in Addition to Other Pathways*

Since we demonstrated that TLK1-AKTIP signaling controls AKT phosphorylation on its regulatory sites (T308 and S473) and the disruption of this signaling either genetically or pharmacologically reduces AKT activating phosphorylation, we examined if the inhibition of TLK1 results in reduced proliferation of the cells. Indeed, LNCaP cells treated with varying concentrations of J54 results in a reduced proliferation rate compared to the vehicle control (DMSO)-treated cells (Figure 4). Unexpectedly, myristoylated AKT overexpressing LNCaP cells (LNCaP Myr-AKT), which are characterized by lipid raft-residing, constitutively active AKT, independent of PI3K pathway activation, can be equally inhibited in their proliferation rate upon TLK1 inhibition, by varying concentrations of J54 (Figure 4). This suggests that in addition to TLK1>AKTIP>AKT signaling, other pathways through TLK1 signaling, such as TLK1>NEK1>YAP, exist and regulate the survival and proliferation of PCa cells, as demonstrated by our previous work [12]. In support of this hypothesis, we monitored NEK1-T141 phosphorylation (the residue uniquely phosphorylated by TLK1) and could demonstrate its time- and dose-dependent inhibition by J54 in both the control and Myr-AKT expressing cells; it is clearly an independent substrate of TLK1 that does not involve active AKT (Figure 4b). We should recall that NIMA was identified in Aspergillus through the loss of function mutations that are unable to proceed through mitosis; that

overexpression of a kinase-dead mutant of NIMA in mammalian cells results in G2 arrest, despite the presence of 11 NEKs, of which NEK1 is the first representative [31]. In previous work with LNCaP cells treated with J54, we reported an accumulation of cells in G2/M [29], and this appears to be the main explanation for their growth arrest, rather than an effect on G1/S progression. In fact, a determination of the PCNA levels in both the LNCaP and Myr-AKT expressing cells did not reveal any differences at early time points following treatment with J54 (Figure 4b). This suggests that J54 does not affect the fraction of DNA replicating cells at early time points, as indicative of a deficiency in G1/S progression, a function that has been shown in some cell models to be dependent on AKT [32]. In addition, an analysis of the ratio of intact vs. cleaved PARP did not reveal an appreciable change in either the control LNCaP or Myr-AKT expressing cells, which would suggest an increase in apoptotic cells at early time points, following the addition of J54. However, it remains possible that at later times (subsequent cell divisions), there would be an increase in the fraction of apoptotic LNCaP cells, compared to Myr-AKT expressing, which we have not investigated. A key requirement for the interpretation of these experiments was the evidence that pAKT would remain unaffected by J54 in the Myr-AKT expressing cells, and in fact, pAKT-S473 was not changed, even at high concentration of J54, in contrast to the control LNCaP (Figure 4b).

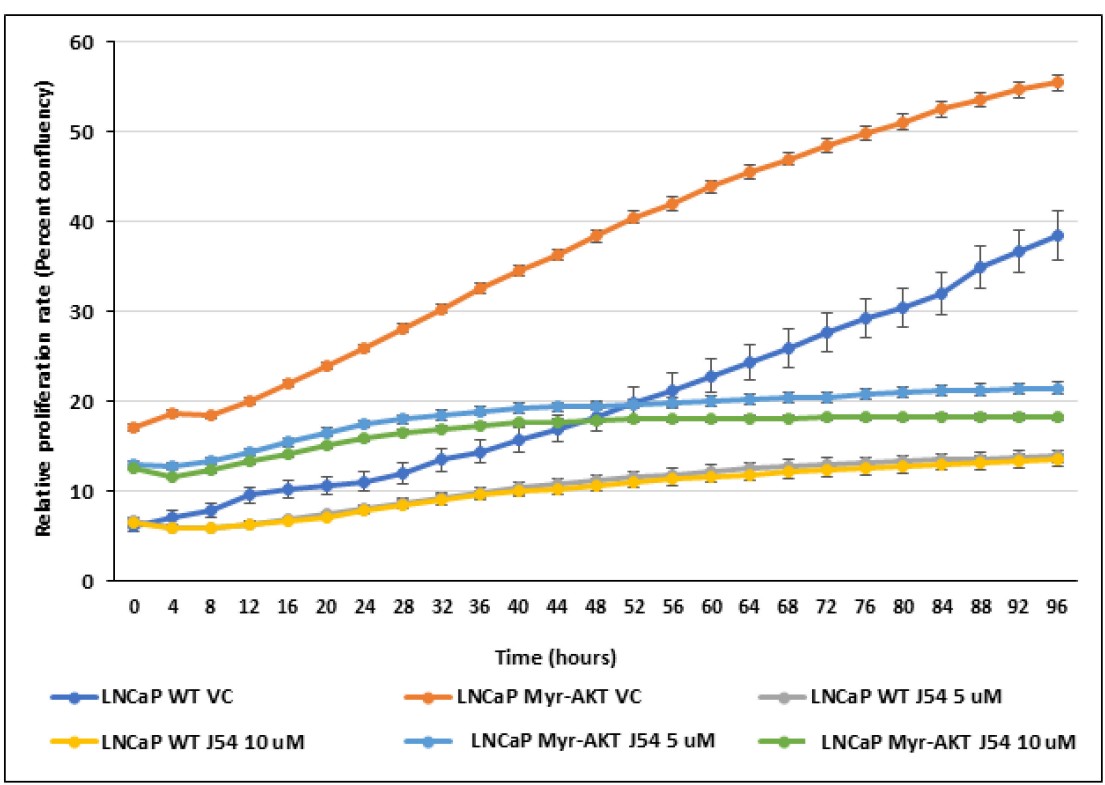

(**a**)

**Figure 4.** *Cont.*

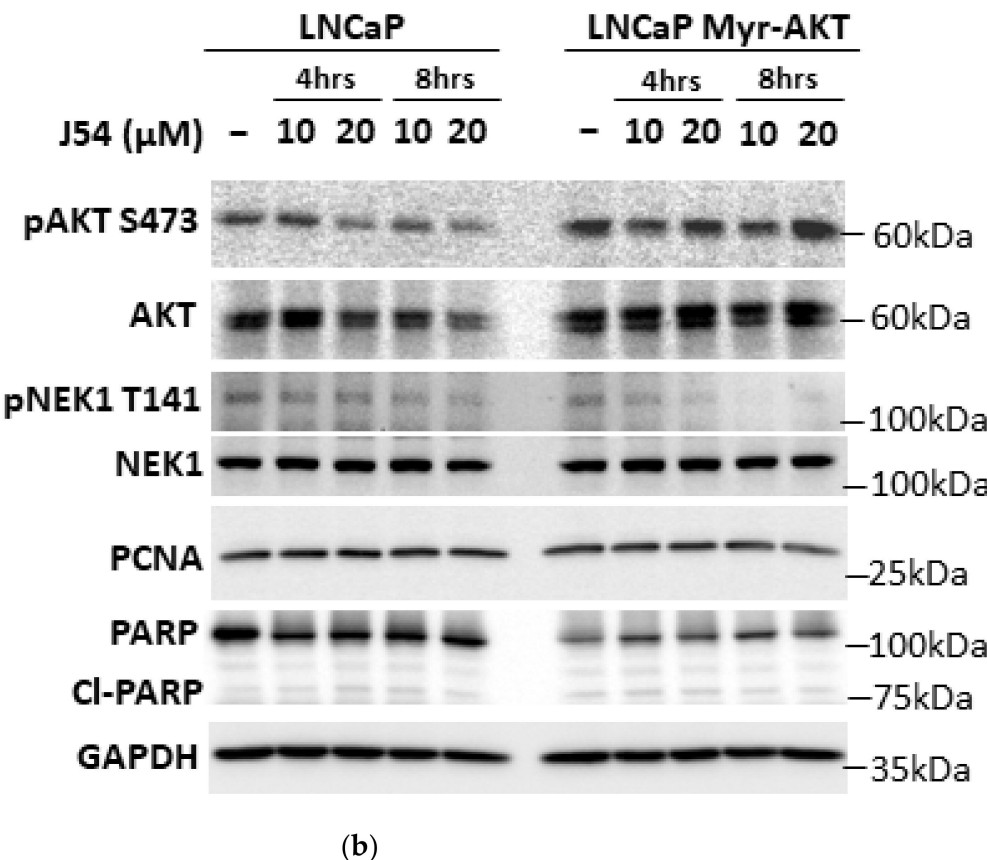

(**b**)

**Figure 4.** TLK1 regulation of cell proliferation is mediated through AKT activation as well as other pathways. (**a**) Proliferation rates of LNCaP and Myr-AKT expressing LNCaP cells with or without two different concentrations of the TLK1 inhibitor J54 treatment. The growth curves with standard error of mean (SEM) were obtained by plotting confluence percentage over time using IncuCyte. (**b**) Cell lysates were prepared from LNCaP and Myr-AKT expressing LNCaP cells treated with two different concentrations of J54 for two different time points. WB was conducted to determine the level of pAKT S473, total AKT, pNEK1 T141, total NEK1, PCNA, PARP, and cleaved PARP. GAPDH was used as a loading control.

## 4. Discussion

The PI3K/AKT pathway has been implicated as a key driver in a variety of tumor types, including prostate cancer (reviewed in [33]). In response to the activation of receptor tyrosine kinases (RTK), the phosphatidylinositol-trisphosphate (PIP3) generated by the activity of PI3K recruits AKT to the membrane via its PH domain, where it is phosphorylated by PDK1 at T308, followed by phosphorylation at S473 by mTORC2 for full activation. Activated AKT contributes to tumor progression in a number of ways, including the enhancement of pro-survival signaling and the inhibition of pro-apoptotic signaling. AKT activation can occur independently of RTK signaling, as a result of the mutation of the tumor suppressor PTEN, a phosphatase that converts PIP3 to PIP2, and is frequently mutated in a high proportion of primary and metastatic tumors, including CRPC. In addition, activating mutations in the catalytic subunit of PI3K or the amplification of upstream RTKs may contribute to enhanced AKT signaling in CRPC.

Importantly, AKT signaling is enhanced under the conditions of an androgen receptor (AR) blockade, driving the resistance to the anti-androgen therapy and enabling the tumor to circumvent the need for androgens to promote tumor cell growth and survival [34,35]. Indeed, clinical studies highlight the critical role of AKT signaling in the emergence of antiandrogen resistance and tumor progression in mCRPC, and a recent phase II study showed that combination of AKT inhibitor ipatasertib (upstream of mTOR and TLK1B) with abiraterone prolongs progression-free survival in mCRPC patients compared to abiraterone

alone [36]. However, the regulation of AKT activation in response to antiandrogen therapy is still incompletely defined. The AKT effector mTORC1 regulates protein synthesis through the phosphorylation of 4EBP-1 and release of the translation initiation factor eIF4E (reviewed in [37]). Prior studies from our group showed that the overexpression of eIF4E could promote the expression of TLK1B [38,39]. Interestingly, we showed that androgen deprivation in the androgen-sensitive cell line LNCaP also increased TLK1B expression [10]. In that study, we also showed that the LNCaP xenografts in mice treated with bicalutamide displayed increased AKT phosphorylation, consistent with the emergence of survival signaling under the conditions of an AR blockade. Thus, elevated TLK1B is associated with an increase in activated AKT, following an androgen deprivation treatment. In a search for TLK1B effectors that might explain the increase in AKT activation, our group identified an AKT-interacting protein (AKTIP) as a putative interactor of TLK1B that could mediate enhanced AKT activation. AKTIP was previously shown to interact with AKT and increase AKT phosphorylation and activity by enhancing its association with PDK1 [16]. In this study, we explore the functional interaction between TLK1B, AKTIP, and AKT and its significance for the regulation of AKT activity. We first confirmed the TLK1 binding of AKTIP, both in vitro and in vivo, and determined the phospho-acceptor sites of AKTIP by TLK1 (Figures 1 and 2). The reduction of activating phosphorylation in the AKT regulatory positions (T308 and S473) was demonstrated by the pharmacologic inhibition and genetic depletion of TLK1 and AKTIP, which suggests that the TLK1 phosphorylation of AKTIP may regulate AKT activity (Figure 3). However, it should be noted that the pharmacologic inhibition of TLK1 did not appear to affect the interaction of AKTIP with AKT (Figure 1d), and as such, the more likely explanation for the functional phosphorylation of AKTIP is to enhance the 3-way interaction with PDK1, resulting in the increased phosphorylation of AKT at the activation sites. Finally, the observation that the proliferation rate of both wild-type and constitutively active AKT in LNCaP cells are reduced upon TLK1 inhibition implies that other pathways for cell survival regulated by TLK1 exist, along with the TLK1>AKTIP>AKT signaling axis (Figure 4). It is noteworthy that inhibition of TLK1 with J54 did not result in the reduced phosphorylation of AKT-S473 in the Myr-AKT expressing cells (Figure 4b).

The TLK1 phosphorylation sites of AKTIP lie in the intrinsically disordered regions (IDR) of the protein (T22 = N-terminal IDR, S237 = C-terminal IDR, see Figure 2f). Phosphorylation in the IDR region may bring about a conformational change and increase the affinity of AKTIP, to better act as a scaffold for the interaction of AKT and PDK1. It is noteworthy that AKTIP uses its C-terminal domain to interact physically with both AKT and PDK1 through their non-catalytic C-terminal domains and to colocalize to plasma membranes [16]. It is also conceivable that the N-terminal IDR phosphorylation of AKTIP may recruit it to the plasma membrane, while the C-terminal IDR phosphorylation increases its scaffolding affinity for AKT and PDK1, thereby recruiting them together to the PIP3 binding site and increasing the efficiency of AKT phosphorylation by PDK1 on T308, followed by S473 phosphorylation by mTORC2. Fully active AKT can phosphorylate mTORC1, which regulates the translation of TLK1B and suggests a possible feed-forward loop. Activated AKT can phosphorylate BAD kinase (S112 and S136), cell death protease caspase 9 (S196), and transcriptional activator FoxO3 (T32, S253, and S315) and suppress their pro-apoptotic functions, thereby mediating CRPC progression (See Figure 5) [40–42].

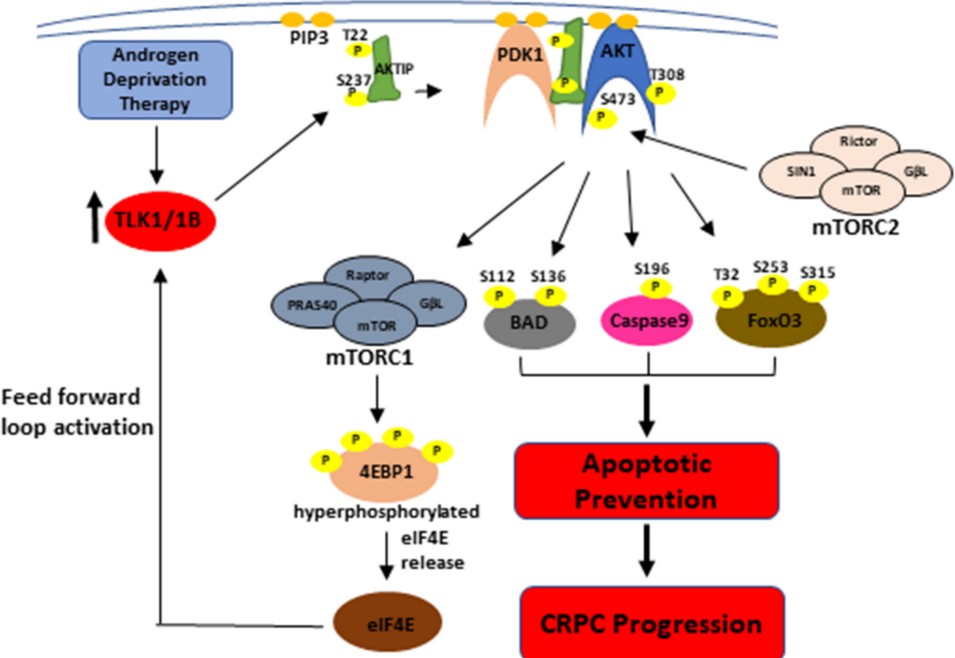

**Figure 5.** Graphical abstract depicting the regulation of AKT activation by TLK1-AKTIP signaling, and thereby, CRPC progression of PCa cells by apoptotic prevention. TLK1 phosphorylation of AKTIP increases AKT phosphorylation at its regulatory sites. AKT phosphorylates BAD, Caspase9, and FoxO3 and inhibits their pro-apoptotic function. AKT can also activate mTORC1 which will phosphorylate 4EBP1 and release eIF4E. Excess eIF4E will increase the translation of TLK1B isoform, and hence, initiate a positive feed-forward loop.

## 5. Conclusions

There has been an ongoing search for the discovery of novel targets to treat CRPC; targeting kinases has been a top priority, due to their druggability and efficiency in treating cancer. Therefore, targeting TLK1, in combination with other therapeutic modalities, to disrupt the TLK1>AKTIP>AKT pathway and impair AKT activation could sensitize the PCa cells to apoptotic death and inhibit lethal CRPC progression. Inhibiting TLK1 could be an excellent strategy to treat patients who have amplification of TLK1 expression or do not respond to PI3K/AKT inhibitors. However, we should caution that inhibiting TLK1 with J54 also resulted in the growth inhibition of the LNCaP cells expressing constitutive Myr-AKT, suggesting that TLK1 mediates other proliferative pathways, such as NEK1>G2/M progression or NEK1>YAP, and is, therefore, not highly specific to the activation of AKT. In practical terms for patients' treatment, the use of TLK inhibitors like J54 to impair the TLK1>AKTIP>AKT nexus and enforce a pro-apoptotic response is very appealing, even though we could not detect this at early time points in LNCaP cells treated with J54. However, this should be studied in the context of combination treatment with other therapy like docetaxel/cabazitaxel [43] or ADT [29].

**Supplementary Materials:** The following are available online at https://www.mdpi.com/article/10.3390/pathophysiology28030023/s1, Figure S1: TLK1 and AKTIP interaction. (A) In vitro pull-down of recombinant His-TLK1B and GST-AKTIP isolated on Ni-NTA Sepharose and immunoblotting of AKTIP. (B) Co-IP of TLK1 from HEK 293 and LNCaP cell lysates using TLK1 specific antibody and immunoblotting for both TLK1/1B (upper panel) and AKTIP (lower panel).

**Author Contributions:** A.D.B. conceptualized and supervised the project. C.M., M.I.K., and I.G. performed all the experiments. M.I.K. and C.M. prepared the figures and quantified the data. M.I.K., A.D.B., and R.M.A. contributed to the writing, revision, and editing of the manuscript. Funding was secured by A.D.B. All authors have read and agreed to the published version of the manuscript.

**Funding:** This work was supported by DoD-PCRP grant W81XWH-17-1-0417 to ADB.

**Data Availability Statement:** Further information and requests for resources and reagents should be directed to the corresponding author. All data generated and analyzed during this study are available upon request.

**Acknowledgments:** We like to thank Jing Chen and Haining Zhu from the University of Kentucky Proteomics Core Facility for conducting the mass spectrometric analysis. We would also like to thank the INLET facility staff of LSUHSC-Shreveport for their assistance in working with the IncuCyte. We also like to thank Feist-Weiller Cancer Center (FWCC) for a predoctoral fellowship to MIK.

**Conflicts of Interest:** The authors declare no conflict of interest.

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
