# Peer review of "Interaction of TLK1 and AKTIP as a Potential Regulator of AKT Activation in Castration-Resistant Prostate Cancer Progression"

_pathophysiology, doi:10.3390/pathophysiology28030023_

Round 1

Reviewer 1 Report

Madere and colleagues investigate the interactors of TLK1 and identify AKTIP as a putative binders and phosphorylation target. Along with this the authors try to establish a link between this and AKT along with an functional importance in cell proliferation.

In our opinion the starting point of the work is interesting and has potential interest.

However there are some major caveats.

  • The quality of the protein-protein interaction data is poor and the experiments should be cleaned and repeated, to represent a valid proof of the concepts stated in the paper. Namely, the data presented in figures 1 to 3.
  • The proliferation assay (figure 4) is a too superficial analysis to be sufficient to prove a cause-effect relationship as described in the work
  • The conclusive figure 5 and the discussion in the test (in particular lines 319-333) is far too speculative
  • The title of the paper is far too speculative

Author Response

  • The quality of the protein-protein interaction data is poor and the experiments should be cleaned and repeated, to represent a valid proof of the concepts stated in the paper. Namely, the data presented in figures 1 to 3.

We have included in the SI a repeat of the TLK1-AKTIP co-IP of fig. 1C, which actually was a previous attempt, and is a bit overloaded but clearly shows their interaction.  We have also added a repeat of the in vitro interaction of the 2 proteins (of fig.1A) and the full blot of the His-pulldown with cell extract (1B).  In addition, we have added the AKTIP-AKT co-IP (fig.1 D).

  • The proliferation assay (figure 4) is a too superficial analysis to be sufficient to prove a cause-effect relationship as described in the work

I would like to refer the reviewers to the Incucyte description of their proliferation assay, which is by now superior to many of the older assays as it allows for much more frequent monitoring of growth - https://www.essenbioscience.com/en/applications/cell-health-viability/cell-proliferation-assays/  .  In addition, we have added in fig.4B, a PCNA wb for early time points after J54 addition.

  • The conclusive figure 5 and the discussion in the test (in particular lines 319-333) is far too speculative

We have obtained additional data on the effect of THD (TLK inhibitor) on the AKTIP-AKT interaction, and therefore have modified slightly the model, which we hope is now less speculative.

  • The title of the paper is far too speculative

Unfortunately, the Rev. did not make clear what part of the title was speculative.  The role of AKT in PCa progression to CRPC is well established, and the role of TLK1 in the same process was clearly proven in several recent publications from our group.  That AKTIP appears to be a regulator of AKT was also previously established by others…so it is not clear what remains speculative.

Reviewer 2 Report

In the current study, authors tried to demonstrate AKTIP to be the interacting protein of TLK1B. Upon interaction, AKTIP was phosphorylated at T22 and S237. Inactivation of TLK1B led to decrease of AKT phosphorylation and led to decrease of LNCaP proliferation. The following questions should be addressed.

  1. In fig1A, B, and C, please indicate the sizes of targeted protein. In Fig1B, please indicate the samples with clear labeling, what does “Column” indicate? According to the manuscript, authors were using immunoblotting to assay the TLK1B interacting protein in elution from Ni-NTA Sepharose pull down assay with HEK293 cell extract, what are the protein sizes of TLK1B and AKTIP. In the same immunoblot, please show protein marker, and whole membrane image, and explain each lane in Fig1B in much more details. In Fig1C, please show protein marker and the immunoblotting result of inputs of TLK1 in both LNCaP and HEK293 cell lysates.

  1. Fig 2 D,E,F numberings are inconsistent from the text and figure legends.

  1. Fig3, it is hard to make any quantitative conclusion on phospho-AKT level without total-AKT expression level. Please show total-AKT level from the same loading for Fig3A,C,G. And, samples need to be on the same blot to show the siAKTIP effect on AKT phosphorylation at T308 in Fig3A. In Fig3E, in THD treated cells showed the total-AKT level also decreased, therefore, it is hard to conclude that pAKT(S473) level is dose dependently decreased by THD treatment. Please show evidence J54 is a specific inhibitor to TLK1. Please label the blot on the bottom of Fig3E, it is over-exposed and cannot be used as part of result. Fig3G, clearly beta-tubulin showed very uneven loading, it is hard to conclude anything.

Author Response

In the current study, authors tried to demonstrate AKTIP to be the interacting protein of TLK1B. Upon interaction, AKTIP was phosphorylated at T22 and S237. Inactivation of TLK1B led to decrease of AKT phosphorylation and led to decrease of LNCaP proliferation. The following questions should be addressed.

  1. In fig1A, B, and C, please indicate the sizes of targeted protein. In Fig1B, please indicate the samples with clear labeling, what does “Column” indicate? We have replaced 1B with the full blot.  The column is Ni-Sepharose which binds His-TLK1B added to the extract.

According to the manuscript, authors were using immunoblotting to assay the TLK1B interacting protein in elution from Ni-NTA Sepharose pull down assay with HEK293 cell extract, what are the protein sizes of TLK1B and AKTIP. In the same immunoblot, please show protein marker, and whole membrane image, and explain each lane in Fig1B in much more details. In Fig1C, please show protein marker and the immunoblotting result of inputs of TLK1 in both LNCaP and HEK293 cell lysates. We have added these, and included in the SI a replica of the experiment with also input lanes, and also have added protein size markers.

  1. Fig 2 D,E,F numberings are inconsistent from the text and figure legends. We don’t think we found inconsistencies.

  • Fig3, it is hard to make any quantitative conclusion on phospho-AKT level without total-AKT expression level. Please show total-AKT level from the same loading for Fig3A,C,G. And, samples need to be on the same blot to show the siAKTIP effect on AKT phosphorylation at T308 in Fig3A. We were not able to reprobe all the blots we had, so we analyzed the samples again in additional blots to determine pAKT, where indicated. In Fig3E, in THD treated cells showed the total-AKT level also decreased, therefore, it is hard to conclude that pAKT(S473) level is dose dependently decreased by THD treatment. THD or J54 did not affect the level of AKT (we have shown this now in Fig. 3).  This was verified in numerous repeats and is now shown in Fig.1A.  Please show evidence J54 is a specific inhibitor to TLK1 This is fully described in DOI: 10.1016/j.isci.2020.101474

Please label the blot on the bottom of Fig3E, it is over-exposed and cannot be used as part of result We have replaced that bottom blot. Fig3G, clearly beta-tubulin showed very uneven loading, it is hard to conclude anything.  We have included AKT blots for all those panels, and also a GAPDH blot which shows even loading instead of the tubulin blot.

Round 2

Reviewer 1 Report

We recognize that the authors made an effort to reply to our objections. We still have some hesitation concerning the suggested link between cancer and the molecular events described. We would suggest modifying the text - where possible- to express caution about the real significance of the data for cancer patients.